# Antinuclear Autoantibodies in Health: Autoimmunity Is Not a Synonym of Autoimmune Disease

**DOI:** 10.3390/antib10010009

**Published:** 2021-02-25

**Authors:** Irina A. Pashnina, Irina M. Krivolapova, Tamara V. Fedotkina, Varvara A. Ryabkova, Margarita V. Chereshneva, Leonid P. Churilov, Valeriy A. Chereshnev

**Affiliations:** 1Regional Children’s Clinical Hospital, 620149 Yekaterinburg, Russia; irisha773@mail.ru; 2Institute of Immunology and Physiology of the Ural Branch of the Russian Academy of Sciences, 620049 Yekaterinburg, Russia; mchereshneva@mail.ru (M.V.C.); v.chereshnev@mail.ru (V.A.C.); 3Laboratory of the Mosaics of Autoimmunity, Saint Petersburg State University, 199034 Saint Petersburg, Russia; t.v.fedotkina@gmail.com (T.V.F.); varvara-ryabkova@yandex.ru (V.A.R.); elpach@mail.ru (L.P.C.); 4Saint Petersburg Research Institute of Phthisiopulmonology, 191036 Saint Petersburg, Russia

**Keywords:** autoimmune diseases, antinuclear antibodies, antinuclear factor, functional autoantibodies, natural autoantibodies, physiological autoimmunity

## Abstract

The incidence of autoimmune diseases is increasing. Antinuclear antibody (ANA) testing is a critical tool for their diagnosis. However, ANA prevalence in healthy persons has increased over the last decades, especially among young people. ANA in health occurs in low concentrations, with a prevalence up to 50% in some populations, which demands a cutoff revision. This review deals with the origin and probable physiological or compensatory function of ANA in health, according to the concept of immunological clearance, theory of autoimmune regulation of cell functions, and the concept of functional autoantibodies. Considering ANA titers ≤1:320 as a serological marker of autoimmune diseases seems inappropriate. The role of anti-DFS70/LEDGFp75 autoantibodies is highlighted as a possible anti-risk biomarker for autoimmune rheumatic disorders. ANA prevalence in health is different in various regions due to several underlying causes discussed in the review, all influencing additive combinations according to the concept of the mosaic of autoimmunity. Not only are titers, but also HEp-2 IFA) staining patterns, such as AC-2, important. Accepting autoantibodies as a kind of bioregulator, not only the upper, but also the lower borders of their normal range should be determined; not only their excess, but also a lack of them or “autoimmunodeficiency” could be the reason for disorders.

## 1. Introduction

The incidence of autoimmune diseases (ADs) is high worldwide among both adults and children. According to various estimates, the total incidence of ADs in different countries and regions varies from 5% to almost 30%, and there is an annual increase [1,2,3,4,5]. This is particularly the case for systemic ADs with non-organ-specific autoantibodies [1]. ADs significantly impair the quality of life of patients and often lead to severe, usually lifelong disability. They require significant costs from the health care system when diagnosed at late stages: for example, the annual costs of treating one case of systemic lupus erythematosus (SLE) with renal or neuropsychiatric complications in the USA in 2013 exceeded 30,000–32,000 US $, being 6.25–6.5 times higher than the cost of treating the initial, inactive, and uncomplicated phases of the disease [6]. Early diagnosis of ADs is desirable, but the process often takes a long time, due to the absence of specific symptoms in early stages of these diseases in many patients [7,8,9,10]. This factor determines the relevance of the search for laboratory tests suitable for the early diagnosis and screening of ADs in order to timely prescribe appropriate treatment.

Identification of different autoantibodies is used for the diagnosis of ADs [11,12]. Some of them are associated with specific autoimmune diseases [10,13,14,15]. However, the reliability of autoantibodies as pathognomonic markers of a particular disease is far from 100%; moreover, many of them are observed in several different nosological entities. Their association with a certain disease sometimes requires confirmation of antigenic specificity by several different methods; therefore, the general consensus is that the term “disease associations” should be replaced by “clinical relevance” of the identified autoantibodies [16].

In addition, there is a growing body of evidence that the responsibility of the immune system is not only (and even not mainly) the protection against foreign intrusions. The immune system serves as a physiological sensory and analytical instrument in relation to the antigenic structure of the multicellular organism, responsible for its maintenance and even its formation. Therefore, auto-recognition is regarded as a normal primary function of the immune system, which is associated with the existence of physiological autoimmunity [17,18].

In this review, we address the issue of antinuclear antibody (ANA) testing, which has been detected by indirect immunofluorescence and known since 1958 under the traditional name “antinuclear factor” (ANF) [19]. We discuss the ANA presence and role in healthy individuals, the peculiarities of distinguishing between normal and pathological positive ANA-test results, the role of ontogenetic and geographical factors in ANA prevalence—in the context of the concept of physiological autoimmunity—and its relationship with ADs.

## 2. Physiological Autoimmunity and Its Bidirectional Pathological Changes

Almost all autoantibodies, including ANA, are often found not only in the sera of patients who suffer from ADs, but also in healthy persons (including those who do not develop a disease during follow-up), although in health, low titers of autoantibodies are usually found [19,20,21,22].

The issue of natural autoantibodies and physiological autoimmunity has acquired considerable relevance with the development of more sensitive laboratory tests, because autoantibodies to many antigens, including cell nuclei and membrane receptors, have become routinely registered in the blood and mucous secretions of healthy people with these methods [23,24,25,26,27,28,29]. This is also true for the autoantibodies that are considered typical markers of certain ADs, for example, IgG against the glomerular basement membrane, proteinase 3, myeloperoxidase, myelin basic protein, etc. [30,31].

Moreover, due to the presence of both agonistic and antagonistic effects of such autoantibodies of healthy donors on the receptors of neurotransmitters [32,33], hormones [34,35], autacoids [36,37], and IgE [38], the question of their possible physiological regulatory role has long been raised. Indeed, there is growing evidence for this aspect of physiological autoimmunity [39,40,41,42].

### 2.1. Physiological Autoimmunity: Historical Perspective and Contemporary Understanding

In accordance with the assumption of the possible regulatory role of autoimmunity, several fruitful scientific doctrines have been formulated in different periods, rooted in the ideas of I.I. Mechnikov, who considered the immune system as a means of forming and maintaining the multicellularity of the metazoan organism [43]. These doctrines are: the doctrine of cytotoxins, the theory of autoimmune regulation of cellular functions, and the theory of immunological clearance [39,41,44,45]. The term “functional autoantibodies” (which primarily relates to the autoantibodies against plasma membrane receptors) was coined in recent years and indicates the modern version of the mentioned ideas [32,46,47].

However, with regard to the functional properties, ANAs cannot be considered an exception, since it was shown in vivo and in vitro that ANAs can penetrate not only into the cytoplasm, but also into the nuclei of living cells (moreover, this occurs with the involvement of antigen binding sites). ANAs can influence gene expression, cell growth, and apoptosis. Therefore, not only regulation, mediated by membrane receptors, but also repression/activation of genes through cis-regulatory elements of chromatin can be carried out by autoimmune mechanisms [44,48,49,50,51,52].

It is no coincidence that there is a growing number of studies which showed that:Patients with certain ADs show a decrease in the level of certain autoantibodies and/or the strength of antibody-mediated bioeffects in comparison with healthy donors;The level of autoantibodies decreases, rather than increases during exacerbations of some ADs;Some autoantibodies are associated with a favorable outcome of the disease.

This was demonstrated both for the autoantibodies to cell surface receptors [42,53,54,55,56] and for ANA [12,57]. For example, anti-DFS70 autoantibodies against the lens epithelium-derived growth factor are more prevalent in healthy people than in patients with ADs, and have been considered in recent years as an important marker of the lower probability of rheumatic diseases, even when the ANA test is positive [58]. However, recent findings raise some questions regarding the clinical relevance of anti-DFS70 autoantibodies. Zheng et al. showed that the frequency of systemic autoimmune rheumatic diseases in anti-DFS70 positive pediatric patients was unexpectedly as high as 50.0% [59]. Conticini et al. reported that 7/9 (77.8%) anti-DFS70 positive adult patients with clinical suspicion of primary Sjogren syndrome received this diagnosis after minor salivary gland biopsy [60].

The paradigm of “beneficial autoimmunity” [18,45,61] is also supported by the use of autoantibodies against nuclear antigens obtained from patients with ADs for the treatment of certain cancers [42,50].

Traditionally, it was assumed in diagnostic immunology that the more autoantibodies (either damaging or functional ones) a patient has, the more symptoms of the disease that will be present [62].

However, apparently, the level of some autoantibodies in a healthy individual should be no more, but no less than the optimum [53,63]. It has been suggested, by analogy with endocrine disorders which can equally occur both from excess and deficiency of a certain signaling molecule, that not only an increase, but also a pathological decrease in the concentration of autoantibodies may reflect and even cause pathological processes in the body [64]. Thus, autoimmunity and ADs are not synonymous terms.

Only a pathological intensity of the autoimmune reaction or its insufficient regulation causes illness. Perhaps, along with the terms that have been coined long ago to denote excessive (“allergy”) and insufficient (“immunodeficiency”) response of the adaptive immunity to foreign antigens, we should start using the terms “autoallergy” and “autoimmunodeficiency” to indicate diseases caused by disorders of the natural self-recognition process being polarized in opposite directions [63,64].

According to the current opinion, ANAs, similar to other autoreactive immunoglobulins and lymphoid clones, belong to the integral part of the normal functioning of the immune system [24,28,32,36,39,40,41,42,44,45,46,47,63,64,65].

Notably, as early as in 1984, N.K. Jerne [66] established the theory that the immunity is controlled and restrained by nothing more than natural autoimmunity, implying an idiotype-anti-idiotypic network of mutually recognizing antibodies and lymphoid clones. Now this theory is complemented with the data on the origin of T- regulators which arise from the differentiation of autoreactive clones with moderate affinity [67].

However, the role of natural autoantibodies (including ANA) in immunity and, more broadly, in the homeostasis of the body is still not entirely clear. Since the 1980s, some studies demonstrated the existence of “natural” autoantibodies. They are produced by B1 cells without antigenic stimulation and are considered the part of the innate immune system [28,65,68]. Their source is the so-called CD20CD27CD43CD5+ (or CD5−) CD70− B cells, a self-sustaining population of descendants of fetal rudimentary lymphoid cells which are generated in the liver and bone marrow in early ontogenesis. They are originally predominant in serous cavities (peritoneal and pleural), capable to settle elsewhere, including the lamina propria of the gastrointestinal tract and inflammation foci, but only occasionally present in encapsulated secondary lymphoid organs [69].

What distinguishes natural autoantibodies is their specificity for a wide range of structurally unrelated antigens such as DNA and insulin, phospholipids and myelin basic protein, oxidized lipoproteins, etc. [70,71,72]. Natural autoantibodies are primarily IgM, but sometimes they belong to other isotypes [42,73,74]. The aforementioned subset of B cells, which is related to the paleo-immunity, undoubtedly plays an important role, in particular in the suppression of the hypersensitivity of the mucous membranes and in the development of oral tolerance [69]. Low titer autoantibodies in healthy organisms, most probably, are products of autoreactive B cells that fail to receive “help” from autoreactive T cells. However, this is just the mechanism of their generation; it does not rule out the possibility of their regulatory effects in health, because these molecules are able to recognize and bind their targets, altering their metabolic destiny and/or lifecycle.

However, we consider that the entire phenomenon of physiological autoimmunity cannot be attributed only to the competence of B1 lymphocytes. After all, autoantibodies from healthy individuals often have high affinity to their targets (for example, anti-DFS70 autoantibodies and autoantibodies to receptors of various bioregulators).

Such natural autoantibodies are involved in the elimination of the body’s own antigens and neoantigens which are formed during cell death or alteration of various biomolecules [22,39,45,61,63,70]. They are credited with protective functions related to the immune clearance of certain antigens, including atherogenic lipoproteins, proteins deposited in neurodegenerative diseases, and other pathogenic autoantibodies [55,65,70,75].

The assumption about the important role of natural autoantibodies and macrophages in immune clearance was first made by the inventor of immunoelectrophoresis, Pierre N. Grabar, more than half a century ago [76,77]. At that time, Mechnikov’s approach to the problems of physiological autoimmunity gave rise to many ideas in the literature (mainly written in Russian and French) regarding the role of natural autoantibodies, including their possible involvement in the regulation of cellular membrane permeability and the intracellular content of macromolecules, in antitumor protection, and even in radioprotection [78,79,80,81].

All these approaches were associated with the indisputable conviction of immunologists prevailing at that time and documented in most authoritative handbooks, that such large molecules as immunoglobulins can perform their functions only outside the cells, in biological fluids or on the cell surface [82].

The idea of P. N. Grabar did not require assumptions that contradicted this dogma and immediately found supporters. It was further developed in the concept of immunochemical homeostasis by I. E. Kovalev [83]. In accordance with the main postulate of this concept, the levels of natural autoantibodies are regulated according to the principle of feedback by the number/availability of molecules of the corresponding autoantigens. Since the levels of expression and secretion into the extracellular space of any cytoplasmic, membrane, nuclear, and other autoantigens differ little in healthy individuals, the serum levels of autoantibodies of corresponding specificity also differ just slightly. However, with the development of any disease, the picture changes, to the extent that the natural dynamics of cell populations are distorted.

From this point of view, it is important not only to measure the absolute levels of certain autoantibodies, but also to compare them with the average levels of these autoantibodies in the population of a particular region. An individual’s autoreactivity (i.e., the spectrum and ratios of autoantibodies of different specificity) should also be taken into account and attention should be paid, first of all, to those autoantibodies that produce positive or negative peaks against the background of the general “landscape”. These principles are reflected in some approach to the immunological screening and immunodiagnostics and continue to evolve [37,41,50,84].

### 2.2. Transition from Physiological Autoimmune Response to Autoimmune Disease

Many targets of naturally occurring autoantibodies, such as DNA, histones, nucleoproteins, and phospholipids, are typical components of apoptotic bodies. In this regard, there is a point of view that one of the main physiological functions of moderate autoimmunity is the elimination of apoptotic debris. Notably, that the same antigens which are abundant among the products of apoptosis (mentioned above, as well as the products of apoptogenic proteases, agonists of cytokines receptors, and chemokines receptors) are also targets of pathological autoantibodies in rheumatic ADs [63,85,86]. It is possible that this group of diseases will someday be termed “autoimmune diseases with autoantibodies against the components of apoptosis”, which we have already proposed [63].

Normally, the products of apoptosis are phagocyted by none antigen-presenting CD68-positive tingible body macrophages (possibly—just by means of physiological natural autoantibodies), but in the case of impaired clearance, a significant part of the debris is engulfed by antigen-presenting cells. The enhanced presentation of apoptotic autoantigens triggers an overly enhanced autoimmune response against those autoantigens (in particular, antinuclear ones), which was noted in individuals predisposed to systemic ADs in contrast to healthy ones [86].

It has been shown that immunization of mice with one of the proteins involved in the clearance of apoptotic bodies (pentraxin-3) led to the emergence of protective autoimmunity and was associated with the delayed development of experimental lupus nephritis [87]. Back in the early 1980s, it was found that the number of B-lymphocytes capable of recognizing double-stranded DNA, one of the main autoantigens among ANA, is the same in the blood of SLE patients and healthy donors [88]. The process of antigen presentation and the effects of T cells on this process can determine, therefore, whether autoimmunity will be maintained within the physiological limits, or whether it will evolve into AD.

Thus, it is possible that natural autoantibodies may precede the appearance of pathological autoantibodies. An increased level of polyreactive B cells was found in patients with ADs [88,89]. Mature naive B cells in patients with rheumatoid arthritis and SLE secrete autoreactive/polyreactive antibodies that can recognize classic autoantigens with low affinity [90]. Under the influence of genetic or environmental factors, these autoreactive/polyreactive mature naive B cells can be differentially activated, resulting in the appearance of B lymphocytes which produce antibodies with high affinity for autoantigens [91]. Breaking self-tolerance can occur not only due to the abnormalities of apoptosis (see above), but also due to the modification of autoantigens during inflammatory, neoplastic, or other damaging processes [64], because of cross-reactivity between foreign antigens and autoantigens and between antigen epitopes and anti-idiotypes [36]. Several factors (e.g., destruction of tissues and inflammation) contribute to the production of co-stimulatory molecules which participate in the interactions between the immune cells thus leading to a more active autoimmune response, but when the tissues are intact, autoantibody titers remain low [92]. ADs can be also triggered by the external influences of adjuvants, adjuvant-like substances, and polyclonal immunostimulants, both of natural (infectious and non-infectious) and anthropogenic origin. This polyetiological additive threshold effect on the individuals predisposed to ADs is known as the “mosaic of autoimmunity” concept [93], and the role of inflammatory autacoids as triggers of the intensification of the autoimmune response is postulated by the “danger hypothesis” [92].

Since autoantibodies can appear long before the clinical manifestations of ADs, they can potentially serve as predictive biomarkers for these diseases. Thus, there are some data that antibodies to cyclic citrullinated peptides appear many years prior to the symptoms of rheumatoid arthritis and are almost always absent in healthy individuals [94]. However, these autoantibodies are characterized by significantly higher sensitivity and specificity than ANA, which makes them much more valuable as predictive biomarkers for the laboratory diagnostics [95,96].

At the same time, although there is some evidence confirming that ANA can also be detected in patients with rheumatic diseases long before their clinical manifestations [97,98,99], ANA-positive patients do not necessarily develop ADs in the follow-up, at least for those diseases which are generally recognized as autoimmune ones [100,101,102].

### 2.3. Functional Autoantibodies

The theory of immunological clearance as the main function of natural autoimmunity has evolved in recent years into the concept of functional autoantibodies and their homeostatic role. It is expounded in several papers cited above [29,38,42] and the most recent [103] publications.

However, the authors of these works, as it was before, still see the homeostatic role of autoimmunity only in the possibility of natural self-correction of certain disorders by such autoantibodies (related to the elimination of autoantigens and/or to the interference in their metabolism). In this interpretation, autoimmunity, albeit “beneficial” in such cases, is still related to the disease. That is, according to this view, autoimmunity is more likely not a normal, but a compensatory phenomenon.

The ideological influence of the famous Paul Ehrlich’s “horror autotoxicus” [104] can be read between the lines of this advanced work on functional autoantibodies. For many years, this postulate prevented the majority of the immunologists not influenced by I.I. Mechnikov’s concepts, from the recognition that autoantibodies can be physiological.

At the same time, there was also a more radical interpretation of the phenomenon of physiological autoimmunity, based on the theory of the autoimmune regulation of cell growth and cell functions. The formation of this theory goes back to the works of A.A. Bogomolets and L.R. Perelman on the effect of small doses of organ-specific antisera (in the terminology of that time—“cytotoxins”)—on target organs. According to the modern interpretation of this theory, autoantibodies act as adaptive bioregulators of cell functions such as neurotransmitters or hormones both in health and disease [39,44,105].

That is, physiological autoantibodies represent specific signals addressed not only to superficial, but also to intracellular receptors, including genomic ones. These signals are the part of the network of idiotype-anti-idiotypic interactions and target not only immune cells, but also other cell types, taking part in the regulation of the cell growth, gene expression, and renewal of cell populations [36,39,44,105,106,107,108].

It was shown in a number of works on the model of endocrine cells (adrenal cortex, adenohypophysis, thyroid gland) that IgG against tissue-specific antigens of the cell nuclei, represented by the complexes of DNA and non-histone proteins, is able to stimulate hormone biosynthesis in these cells in specific ways and influence their proliferation (which results in hyperplasia of targeted organs with prolonged exposure). The effect of cytostimulating IgGs targeting the adrenal cortex was reproduced after hypophysectomy in rats and was able to inhibit atrophy of the adrenal cortex in these animals. These antibodies gave a picture similar to ANA in the reaction of indirect immunofluorescence, and serologically identical immunoglobulins were detected in the serum of intact animals by the Ouchterlony test [39,44,64].

For a long time, the development of this concept was restrained by the prevailing opinion about the inability of antibodies to penetrate into living cells (see above), but the work of the Mexican scientist D. Alarcon-Segovia and Russian authors A.S. Zaichik et al., later confirmed by many other scientists [39,48,49,50,52,89,109], demonstrated the ability of ANAs (both experimentally obtained and isolated from the sera of SLE patients) to penetrate into living cells in vitro and in vivo, and showed the biological effects of such immunoglobulins on various genetically determined processes. This concept is also consistent with the data on the enzymatic activity of some antibodies to nuclear antigens in relation to their target antigens (the concept of abzymes) [110].

Thus, an important aspect of the natural autoantibody functioning is their involvement in immune–neuroendocrine regulation as recognizing, signaling or catalytic molecules.

The protective function of autoantibodies in human ADs, especially regarding natural IgM, is equally important [65]. An inverse correlation was found between the level of IgM autoantibodies and disease activity, severity of lupus nephritis, and cardiac manifestations in SLE [111,112,113], articular lesions in rheumatoid arthritis [75] as well as with the severity of non-rheumatic ADs [70]. It was found that natural IgM autoantibodies are involved in the regulation of IgG reactivity in normal sera by binding and neutralizing them [114]. A similar role is attributed to natural autoantibodies against the IgE receptors—namely, down-regulation of immediate hypersensitivity [38]. It should be noted that pathological processes in immune-dependent diseases are more often mediated by autoantibodies of the IgG isotype, although other isotypes may also contribute [115].

In light of the concepts of physiological autoimmunity and natural autoantibodies, it can be expected that not only the upper, but also the lower cut-off levels of some autoantibodies may be diagnostically significant, and protective autoantibodies (including ones against nuclear antigens) will be found to contribute not to the disease, but to the homeostasis. This has been recently shown for anti-DFS-70 autoantibodies in rheumatic ADs [12,16,57,58].

## 3. ANA: Detection, Polyspecificity, and Relation to AD Pathogenesis

ANAs are found in the sera of more than 90% of patients with systemic ADs and can also be detected in many other autoimmune, infectious, and oncological diseases [5,116,117,118,119,120,121]. ANAs represent a wide family of autoantibodies of various specificities that bind to nucleic acids and associated nuclear proteins [115,122,123].

ANF was detected in 1957 in the sera of patients with SLE [122]. Since then, it has become one of the diagnostic criteria for SLE [7,124,125], as well as for other systemic ADs with non-organ-specific autoantibodies: systemic scleroderma [14,120,121], Sjogren’s syndrome, and mixed connective tissue disease [7,23]. ANAs are also present in rheumatoid arthritis [23], autoimmune hepatitis [126], pernicious anemia associated with primary autoimmune atrophic gastritis [127], Hashimoto’s thyroiditis, and immune thrombocytopenic purpura [128].

Certain nuclear staining patterns for ANAs have been described as clinically significant also in dermatomyositis, autoimmune myopathies, primary biliary cirrhosis, Crohn’s disease, antiphospholipid syndrome, autoimmune cytopenias, and occasionally among paraneoplastic phenomena [16].

Several new associations have been revealed between the presence of ANAs and diseases which are not generally considered to be related to autoimmunity: for example, ANAs have been found in idiopathic epilepsy [129], ischemic brain disease [130], interstitial lung diseases [131], schizophrenia [132], and other ailments. According to O.V. Danilenko et al. (2020), the level of autoantibodies to double-stranded DNA belonging to the ANA group is increased in various forms of chronic fatigue syndrome (myalgic encephalomyelitis), especially, when the disease manifests after herpes virus infections [133].

ANF is a historical term introduced in 1960 by the British rheumatologist Eric John Holborow (1918–2009) [134] which was used to characterize the totality of antinuclear antibodies of different specificity, detected by indirect immunofluorescence assay (IFA). IFA is the “gold standard” for ANA detection. Until the mid-1980s, it was carried out on frozen sections of internal organs of animals or humans (spleen, kidney, liver, etc.), which resulted in additional variability of the test results and hindered interlaboratory comparison. In 1983, a standardized object for the ANA-IFA test was proposed [135], namely, HEp-2 cells, which had been cultured by that time for about 30 years in the laboratory as a human laryngeal epithelioma strain [136,137]. The reaction of indirect immunofluorescence with patient sera and fluorescently labeled heterologous antibodies against human immunoglobulins of one or another isotype is carried out according to the standard technique and the result is viewed with a fluorescence microscope. The serum titer and the fluorescence pattern are assessed [138]. Since 2003–2004, solid-phase ANA testing using multiplex fluorescence immunoassay and similar solid-phase methods began to gain popularity as an alternative to IFA. However, IFA on HEp-2 cells, which serve as a kind of natural “microplates or even nanoplates” with a set of autoantigens (circa 100 of them), is still recognized as the most sensitive (and, importantly, visual) method for ANA detection, despite the improvement of the solid-phase immunoassay methods [16,138,139].

When lysates of HEp-2 cells are used as complex antigens in solid-phase immunoassay methods, minor antigenic specificities present in natural cells remain underrepresented, and are therefore not detected. In addition, antigens are present on cells in the native conformation and among natural microenvironments, while other epitopes can be exposed in solution and on a solid-phase carrier. Therefore, except for cost reduction and increased productivity, solid-phase methods, from our point of view, do not provide any other indisputable advantages over IFA.

The use of standardized HEp-2 cells as a substrate for IFA makes it possible to describe various fluorescence patterns, which reflect the presence of immunoglobulins with different antigenic specificity. Each of the fluorescence patterns (to date, the International consensus on ANA patterns (ICAP) describes 29 such patterns) is clinically significant for certain ADs [16]. Detection of ANAs with a description of the type of fluorescence is an important step for the selection of solid-phase immunoassay methods (immunoblotting, enzyme immunoassay, multiplex bead immunoassay, etc.), which are applied to determine antigen specificity.

However, the IFA pattern does not always coincide with the results of the solid-phase assay [140]. In addition, IFA is a more laborious and time-consuming technique in comparison with fully automated biochemical tests, and the visual interpretation of the results could be subjective. Moreover, there are IFA patterns for which the clinical significance has not been unequivocally established.

Being relevant for the diagnosis and prognosis of mixed connective tissue disease and systemic sclerosis and even criterial for diagnosis of SLE (with sensitivities 90–95%), the HEp-2 ANA test is only helpful (with 45–80% sensitivities) for the diagnosis of a few other autoimmune diseases (autoimmune hepatitis, dermatomyositis/polymyositis, Sjogren’s syndrome), and irrelevant in the iagnosis of Hashimoto’ s disease or rheumatoid arthritis (due to sensitivities of only 10–20%) [139]. Therefore, according to a survey conducted among the laboratories, only about 50% ANA tests in the USA and up to 75% in other countries as of 2020 were performed by indirect immunofluorescence [141].

This survey showed that genetically engineered HEp-2000 cells with the overexpression of the SS-A/Ro autoantigen (underrepresented in original HEp-2 cell culture) are becoming more common as a substrate for the determination of ANA staining patterns and are already used by about 24% of the laboratories in the USA, but only by about 3% in the other countries. The establishment of automated platforms for the reading of the IFA results to a certain degree made it possible to overcome the subjectivity of the method during the last decade. The agreement between the results of manual and automated HEp-2 ANA tests reached 92–99%, although the hardships are still great in recognition of mixed patterns, and an automated test is combined with manual reading, with still imperfect interobserver agreement [139]. Hence, there are persisting doubts if ANAs revealed by a single HEp-2 IFA-test may serve as an entry criterion even for SLE, and therefore a recommendation was made to use combined IFA and solid-phase assay data [142,143].

In 2020, about 33% of the laboratories used an automated platform for slide preparation; 16% captured images by the automated platform, but only 5% used automation for the interpretation of the images [141]. Modern consensus guidelines for the interpretation of the results of ANA testing by IFA [16], known as ICAP, developed in 2014–2018, have become an important step towards standardization of the interpretation and unification of the nomenclature in this area. Interestingly, in ICAP workshops, it was agreed that regarding the 29 defined fluorescence patterns, the term “disease associations” should be replaced by “clinical relevance”.

With the new knowledge and experience gained in the usage of modern modifications of the old ANA test, it became clear that not only the classic term “ANF”, but also the other ubiquitous one “ANA test” does not fully reflect the variety of IFA data. The fact is that IFA with HEp2 cells allows us to register not only 15 nuclear staining patterns, but also nine cytoplasmic ones, as well as five other ones associated with mitosis, where not only chromosomal autoantigens, but also autoantigens related to the cytoskeleton are involved [16]. The most radical proposal was to rename the detection of ANA by IFA to “a test for anticellular autoantibodies” [144]. However, later a more precise name was suggested: “the HEp-2 IFA” [145,146].

Of course, one can recall in this connection the famous: “A rose by any other name would smell as sweet...”—from the mouth of Shakespeare’s Juliet [147]. However, in this case, the name of the test is an important detail, because different laboratories, reporting to the customer the test result, interpret the detection of cytoplasmic fluorescence patterns during the ANA test in different ways—sometimes as “ANA negative”, although its clinical relevance is obvious [16,141,145].

## 4. Detection of ANAs in Healthy Individuals

There is ample evidence in the literature that ANAs can be detected in healthy subjects by both IFA and biochemical immunoassay methods [148,149,150,151,152,153]. Autoantibodies to nuclear proteins are normally present in the sera of healthy people and intact animals [154,155]. Moreover, in healthy individuals, autoantibodies to double-stranded DNA and their anti-idiotypes are found in blood plasma [156]. Some of the autoantibodies of DNA in the sera of healthy donors are masked by complexes with serum poly-anionic proteins and can be detected after special sample processing, which indicates their wider prevalence under normal conditions than routine laboratory methods show [157]. Therefore, it is especially important to distinguish between normal and pathological levels of autoantibodies for the diagnosis of ADs, also when the HEp-2 IFA test is performed.

### 4.1. HEp-2 IFA Cut-Off Titers

The main disadvantage of the HEp-2 IFA test is its quite low specificity due to the presence of ANAs in the sera of healthy donors. In most published sources, including methodological guidelines, only the upper limit of a normal ANA titer is indicated, since their complete absence is considered the most frequent normal variant (although not the only one). In view of the above (see Section 1 and Section 2), this approach appears insufficient and not up to date.

Autoimmunology is moving towards the recognition of the importance of the normal range of the levels of several autoantibodies, as it has long been accepted for hormonal and any other bioregulators.

In the literature, different values are mentioned as a cut-off from which the HEp-2 IFA test should be considered positive: 1/40–1/80 [158], 1/80 [152], 1/100 [117], 1/160 [159,160], and 1/200 [161,162]. Therefore, the frequency of ANA-positiveness in different studies will be different depending on the chosen cutoff. According to a laboratory practice survey published in 2020 [141], 50% of laboratories in the world accepted a 1/40 titer as a cutoff.

In a review article by Saikia et al., the importance of the ANA cutoff titer is discussed [163]. The authors showed that with 1/40 serum dilution, about 20–30% of clinically healthy people had a positive result. When a 1/80 titer was used as a cutoff, this share reduced to 10–12%; when 1/160 and 1/320 titers were used—it decreased to 5% and 3%, respectively. A similar picture is observed when comparing any data of the authors who indicate in their articles the prevalence of ANA in different titers (Table 1).

For example, in a Brazilian study of 500 healthy adults, 22.6% of samples were found to be positive for ANAs with 1/40 as a diagnostic titer [97]. However, at 1/80, 1/160, and 1/320 dilutions, the rates of positive results were much lower. Other authors from Brazil [164] did not use the 1/40 dilution in their study; therefore, ANA prevalence among their patients was almost two times lower than in the study by Fernandez et al. (Table 1). Moreover, it was [164] who pointed out the connection between the special pattern of fluorescence associated with autoantibodies targeting the DFS-70 antigen and the absence of rheumatic diseases.

As early as 1997, Tan et al. obtained similar results on the optimal ANA titer cutoff. [165]. They concluded that the 1/40 titer includes almost all patients (high sensitivity), but also a significant part of healthy individuals (low specificity). At the same time, a titer of 1/160 excludes more than 95% of healthy people, but “does not notice” a significant part of patients. Therefore, laboratories must report the results for both titers. Moreover, according to these authors, the detection of low-affinity ANAs in small dilutions in persons considered healthy is also biologically and clinically significant. An inspection of laboratories determining ANAs, conducted in 2001 in the USA, showed that almost 60% of laboratories used 1/40 as a cutoff titer, 23% used 1/80, and only 14% used a 1/160 dilution [163].

A similar survey was repeated 19 years later on a larger scale, involving not only 942 American laboratories, but also 264 ones from the other countries [141]. It was shown that modern practice in the countries of the Old and New Worlds is very different: While in the USA, a 1/40 titer was used as a cutoff by 73% of laboratories (that is, more than what was registered 19 years ago), experts from other countries were clearly inclined to a stricter criterion—only 41% relied on the presence of ANAs at a titer of 1/40, and the majority of non-American laboratories considered a cut-off titer of 1/80 (44%).

The authors of one of the papers cited above [163] concluded that each laboratory should determine a regional diagnostic titer for the population, and the authors of the other study [141] reinforced the importance of the reporting to the physicians the titer at which the result is registered, as well as the inadmissibility of labeling the ANA test results as negative if only a non-nuclear pattern of fluorescence is detected.

However, the use of different cut-off levels in the laboratories makes the results inconsistent. For example, parallel testing of 26 samples in two independent laboratories revealed a discrepancy in titers in 18 cases. Since 1/40 and 1/20 titers were used as diagnostic in the first and in the second laboratories, respectively, fluorescence at a dilution of 1/20 was not found and therefore reported as negative in the first laboratory. Moreover, the second laboratory regularly tested the sera in dilutions of 1/320 and 1/1280, which also led to differences in the results [148].

Malleson et al. [168,169], based on the results of the analysis of their own data and literature reviews, noted that ANAs are detected in healthy individuals (children and adults) up to a titer of 1/320. The authors conclude that if these antibodies are found in low titers (<1/640) and there are no clinical symptoms, laboratory results should be ignored. Gilbrio et al. also believe that ANAs are not necessarily associated with AD, sometimes even despite high titers [149]. They concluded that an ANA test is required only in individuals with clinical signs of ADs.

However, if one follows the above logic, then the same ANA titers in the presence of clinical symptoms should be considered, and in their absence should be ignored.

The question naturally arises, what to do with atypical, subtle symptoms of ADs?

Further, what is the meaning and medico-economic justification of a laboratory study if it does not verify a clinically reasonable diagnosis? Such a dual diagnostic interpretation of the test results reduces the significance of these autoantibodies.

This conclusion is confirmed by the results of Abeles et al. [151]. According to these authors, the positive predictive value of ANAs for the diagnosis of ADs with the 1/160 cut-off titer was 11.6%. Even with a titer of 1/640, the positive predictive value for rheumatic diseases was low—26.9%, for SLE—6%, and with a titer of 1/1280, it was 38.9% and 5.6%, respectively.

Similar results were obtained by Turkish researchers [100]. Of the 409 examined children with suspected systemic ADs with non-organ-specific autoantibodies who lived in Turkey, 27.6% had positive ANA titers, and only 15% were ultimately diagnosed with ADs. None of the 13 patients with an ANA titer of less than 1/160 had rheumatic diseases. The positive predictive value of ANAs was 16% for any systemic AD and 13% for SLE [100].

Only 1968 (57.3%) of 3432 patients with suspected systemic AD had ANAs at the titer 1/100 or more, and only in 293 (14.9%) from the “positive” 1968 cases was the suspected diagnosis confirmed [101], that is, the applied efficiency of this laboratory test was very low. In a study conducted in Taiwan (China), which involved 355 patients from a rheumatological clinic with positive results of the HEp-2 IFA test, systemic ADs were more common in those with ANA titers of 1/640 or higher than in those who had ANA titers in the range of 1/40–1/320 [128].

In another study of 205 children with rheumatic diseases, ANAs at a titer of 1/20 or more were detected in 67% of cases, but in 494 children with non-rheumatic diseases—in 64% of cases, which completely negates the value of detection of ANAs at low titer for rheumatic ADs [169].

The relevance of ANA testing for the diagnosis of those diseases characterized by the rarity of high titers of autoantibodies, for example, for juvenile idiopathic arthritis, is especially doubtful, according to the literature data [170] and our practice [171]. However, for the diagnosis of SLE, detection of ANAs is more significant, since the disease is usually characterized by higher levels of autoantibodies [171,172]. However, even for SLE, the diagnostic value of ANAs in the studies cited above did not exceed 15% [100,151].

### 4.2. Regional, Social, and Racial-Ethnic Aspects of ANA Prevalence

In different cohorts of healthy individuals, the prevalence of ANAs can vary significantly. Cacciapaglia et al. found that results of the HEp-2 IFA test were positive in 23.7% of healthy Filipinos who migrated to Italy, compared with permanent residents of Italy, in whom the prevalence of ANAs was 8.3% [166]. Probably, different socio-economic living conditions of these cohorts could be one of the reasons for such discrepancy. Migrants often have a lower socio-economic status than the indigenous population, and this can be the reason for the worse health status. In addition, the authors suggested that the prevalence of ANAs in migrants can be related to the influence of the environment in which they lived before moving. There is evidence that rural residents have a higher incidence of ANAs than the urban population, possibly due to exposure to toxic substances used in agriculture [173,174]. It was shown that 282 (42%) of 668 men living in North Carolina and working with pesticides had ANAs in the sera at a dilution of 1/80 or more [175]. The authors of the study concluded that organochlorine compounds could play a role in the increasing level of ANAs, and, over time, in the development of ADs.

Satoh et al. reported that the prevalence of ANAs in different groups of healthy adults (donors, healthcare workers, healthy volunteers, residents of small towns) varies widely from 1.1% to 20%, depending on the occupation and place of residence [21]. Mexican researchers found that when healthy subjects were tested for ANAs, the titers in health care professionals were higher than in healthy donors and even than in the relatives of patients with rheumatic diseases [20]. Health care workers are often enrolled as healthy volunteers giving control sera in comparative studies: for example, according to Tan et al. [165], up to two-thirds of the control sera were received from employees of the universities where the laboratories were located, including 13% from healthy individuals who professionally were directly involved in sera testing.

At the same time, there are data from studies (albeit rather old ones), which show that ANAs to DNA in healthy volunteers enrolled from the laboratory staff were identified more frequently than in the overall healthy population [176,177].

The geographical features of the ANA prevalence are worth mentioning because their prevalence varies in different countries. For example, when 557 healthy volunteers from different countries were tested for ANAs (by HEp-2 IFA), these autoantibodies were found in 45% of Colombians, 38% of residents of Kitava Island in Papua New Guinea, 26% of Mexicans, 12% of Italians and Dutch, and 11% of Israel residents [150]. According to the Italian authors, the detection rate of ANAs by ELISA was 1.3% in a cohort of 149 healthy adults [13]. When the same laboratory method was used to examine 401 healthy residents of Texas (USA), 25% of them have positive results [178]. With the HEp-2 IFA test, the same tendencies can be noted. Authors of the publications from European countries (where, as mentioned above, laboratories are more likely to take higher dilutions of serum as a cutoff) usually report a rather low prevalence of ANAs in healthy populations. For example, ANAs were detected only in 4.9% of 41 healthy residents of Poland (the titer was not indicated) [127]; among blood donors in the Netherlands at the dilution of more than 1:80, the value was about 4%, although lower titers were found in 12.7% of cases [179].

ANA-positivity in the Americas, according to literature, is higher than in Eurasia. Thus, out of 304 healthy Mexicans, 17.9% had ANAs at a titer of 1/80 and more, but only 1.3% of these individuals were positive at a 1/320 titer [20]. In a study performed in the USA which involved 4754 healthy individuals over the age of twelve years old, 12.8% of ethnic Mexicans, 13.7% of Caucasians, and 15.5% of African Americans were positive for ANAs at the titer of 1/80 [21]. In East Beirut (Lebanon), 10,814 healthy individuals were tested; ANAs at a titer of 1/100 were detected in 26.4% of cases, and the prevalence in population aged over seven years has increased 2.5 times since 2008 [167]. However, in a geographically close region of Turkey, ANAs at a titer of 1/100 were detected only in four people out of 507 healthy individuals (0.78%) [23]. In studies conducted in the countries of Indochina and the Far East, several different results of ANA prevalence have been reported. For example, among 100 healthy adults from Thailand, only eight had ANAs (1/80 cut-off titer) [180]. In an extensive study conducted in Baoding City, Hebei Province, China, just over 5.9% of nearly 20,500 healthy donors tested positive for ANAs, mostly women [181]. Among 33 healthy adults living in Japan, two had ANAs at a titer of 1/40, and two more—at titers 1/80–1/160 [182]. In the international study cited above, conducted in 15 laboratories (Europe, USA, Canada, Australia, Japan), positive results in healthy adults at a titer of 1/80 were detected in 13.3% of cases, and at a titer 1/40—in 31.7% [165].

It is interesting to trace the dynamics of the ANA prevalence in the same country (or population group) for long periods of time. Since this requires referring to biobanks collected over many years according to standard rules, such attempts are rare [167]. A recent research project from the USA which used a biobank, created during the National Survey of Health and Nutrition of Americans, is outstanding because of its scope. Sera of 14,211 US residents over 12 years of age collected over three time periods (1988–1991, 1999–2004, and 2011–2012) were routinely tested for ANAs in one laboratory by HEp-2 IFA, and the results were correlated with medical history and questionnaires completed by the participants [183]. Findings indicate that there is a clear, statistically significant trend towards an increase in the number of seropositive Americans over the years, especially in the most recent time period (from 11% and 11.5% to 15.5%). ANAs are more common in women (20.1%) than in men (11.4%), in those over 50 (20.5%) than in young people (13%), and in African-Americans (18, 1%) than in other racial groups. It does not show a correlation with body mass index (although over the observation period, BMI increased, as did the consumption of alcohol and tobacco products). Contrary to expectations, ANAs are somewhat less likely to be present in active smokers (13.1%) than among non-smokers (17.1%), and are more often found among teetotalers (21.3%) than among those who consume alcohol moderately or frequently (14.8%). The recent increase in ANA seropositivity is most pronounced among white men (from 10.2 to 16.4%) and is especially significant among adolescents (from 5% at the turn of the 1980s–1990s—to 12.8% in 2011–2012 years). Discussing these data, the authors vaguely mention the role of factors acting during gestation and in early life.

However, we consider that the most significant change in early life that occurred between 1989 and 2012 in this case is the establishment of more intense national immunization programs and the adjuvant load associated with these programs, as well as increasing intensity of other anthropogenic adjuvant factors. In 1988, in the United States, only 2.9% among 241 healthy children and adolescents aged four months to 16 years had ANAs at serum dilutions 1/10–1/40 [184], and in 2012, the test turned out to be positive already in 11.2% of the tested American adolescents 12–19 years old, although a higher dilution of serum (1/80) was considered as a cutoff [21].

There are practically no data on the occurrence of ANAs in healthy children and adults living in the largest multinational country of the world—the Russian Federation. We detected ANAs at a titer of 1/80 or higher—in 22 healthy donors out of 100, in the Urals, in the Sverdlovsk region, of which five people had an ANA titer of 1/320, 1–1/640 and 1–1/1280 [185].

Currently, it is common knowledge that geographical differences exist not only in the ANA prevalence among healthy individuals, but in the regional distribution in ADs. For example, Lerner et al. [1] reported that in the 21st century, the most significant increase in the incidence and prevalence of ADs has occurred in the West and North compared to the East and South. This difference is usually associated with a decrease in the incidence of infectious and parasitic diseases in some regions (according to so-called “hygienic hypothesis”). However, there is also an impact of vitamin D supply under conditions of different latitudinal sunlight exposure and additional factors associated with urbanization [1,150].

It can be pointed out that the above information on the ANA frequencies in healthy inhabitants of the New World, in comparison with Europeans, correlates with the data by Roberts et al. [186], who showed, using 52 million medical observation cards from the period 2010–2016, that the multiracial population of the United States also had a higher prevalence of ADs, than the population of Europe. They identified regional intra-American variations in the prevalence of ADs: SLE was more common in African Americans in the West North Central and South Atlantic regions of the country; multiple sclerosis—in African Americans in the South Atlantic and Pacific regions; rheumatoid arthritis—in Native Americans in the Pacific, West North Central, and Mountain areas. These findings represent more evidence for the role of the genetic factors in the etiology of ADs.

Due to the complexity of comparing data from different laboratories, it is preferable to implement large-scale multicenter studies performed according to a single method with the same test systems.

### 4.3. Ontogenetic Aspects of Autoimmunity to the Nuclear Antigens

Many parameters used in modern laboratory diagnostics have different reference intervals, depending on gender and age. However, methodological guidelines and publications related to AD diagnosis are often devoid of any reference to the differences in the normal values for males and females and for people of different ages [23,152,158,162]. Indeed, there is a general tendency with age to immunosuppression against the background of a chronic systemic excess of pro-inflammatory autacoids, the activation of autoimmune processes in the body, and expansion of their spectrum [187]. However, there are few studies which describe the ontogenetic dynamics of ANA levels, detected by the same laboratory, although it is methodologically difficult to compare the results of different laboratories analyzing this question.

In general, the incidence of ANAs in children is comparable to that in adults. However, in the reviewed publications of authors from Europe, Asia, Australia, North and South America, different age groups were tested and different ANA cutoff titers were used (Table 2).

Some of the authors who analyzed the age-related characteristics of the ANA-positivity concluded that its prevalence increases with age [21,167,168]. According to other researchers, no correlation was found between ANA titers and the age of adult donors [20], at least in the range of 20–60 years [165].

Interesting results have been obtained in the study by Fernandez et al. [97]. The authors showed that among 394 adult donors under 40 years old, ANAs were detected in 21.1% of cases at a titer of 1/40 and higher, and in 106 adults over 40 years old, positive results were observed in 28.3% of cases. At the same time, at a cut-off titer of 1/320, the opposite results were obtained: the detection rate in the younger group was 1.5%, and in the older one 0.9%. The data on the children and adolescents of different ages are presented in the publication by Hilbrio et al. [149]. Among the surveyed age groups: 6 months–5 years, 5–10 years, 10–15 years, and 15–20 years, ANAs were most often detected in children from 5 to 10 years old (Table 2).

Presumably, the most common among young people (30–35 years old) was the anti-DFS70 immunofluorescence pattern of the HEp-2 IFA-test, which is suggested to be protective for rheumatic ADs [57,189]. Finally, a large-scale study from the USA cited above [183] registered a trend towards an increase in the detection of ANAs with age (and especially—over 50 years old) and a flattening of the differences between adolescents and young adults, which were observed among individuals whose samples were taken 30 years ago, but almost disappeared in subsequent periods due to the recently increased frequency of ANA-positiveness in adolescents. The frequency of the detection of ANA increases with age, but non-linearly.

### 4.4. ANA IFA Patterns in Health

The ICAP of 2014–2015 describes 28 types of variants of the picture recorded when determining ANAs by IFA [189]. A modernized version of the ICAP 2018–2019 added to them the 29th pattern, verified by experts [16]. The patterns are denoted by the acronym AC (anti-cell) and numbers [190].

Among all nuclear patterns observed in HEp-2 IIFA analysis, the main ones and minimally mandatory for the identification by any laboratory during routine analysis are homogeneous (AC-1), speckled (subtypes which should be reported by “expert-level” laboratories: AC-2, 4, 5), centromere (AC-3), and nucleolar (expert-level subtypes: AC-8, 9, 10). Among the cytoplasmic patterns, those mandatory for routine description are fibrillar (expert-level subtypes: AC-15, 16, 17), speckled (expert-level subtypes: AC-18, 19, 20, 21), polar/Golgi-like (AC-22), and a pattern of rods and rings (AC-23). Mitotic patterns (AC-24–AC-28) are verified exclusively by the experts.

A homogeneous pattern is characteristic of the presence of antibodies in response to the main components of nucleosomes: double-strained DNA and histones; it is clinically significant for the diagnosis of SLE, juvenile idiopathic arthritis, and chronic autoimmune hepatitis [16,140,190,191]. The speckled type of fluorescence is observed in a wide range of rheumatic diseases (Sjogren’s syndrome, rheumatoid arthritis, juvenile idiopathic arthritis, various forms of lupus erythematosus, systemic sclerosis, dermatomyositis, mixed connective tissue disease, overlap syndromes, and undifferentiated connective tissue disease) and is divided into three expert-level sub-types (Figure 1) [12,16,158,171,172,191].

Regarding ANA IFA patterns in healthy people, the most interesting pattern is AC-2—dense fine speckled (DFS) [12,16] (see below).

ANAs to nucleolar antigens are associated with three nucleolar patterns, which are most clinically relevant for systemic scleroderma, but can be also identified in mixed connective tissue disease, Sjogren’s syndrome, and occasionally present as a paraneoplastic phenomenon) [16,140,158,190].

A distinct pattern AC-29 has been recently described and associated with antibodies to Scl-70 (DNA-topoisomerase I), which could also cause a homogeneous pattern, since the concentration of DNA-topoisomerase is maximal in the nucleoles. The AC-29 pattern is highly significant for systemic sclerosis but can also be found in overlap syndrome between this disease and dermatomyositis [16,146,192].

The centromere pattern is important for cutaneous forms of scleroderma and CREST syndrome; it can manifest itself long before the manifestation of other symptoms and, together with Raynaud’s phenomenon, is prognostically extremely valuable for early diagnosis [16,144,193].

The clinical significance of cytoplasmic and mitotic staining patterns is discussed in detail elsewhere [16,140,143].

When ANAs are detected in health, the speckled pattern is the most common, followed by the homogeneous one. For example, Brazilian authors showed that 78% of healthy children positive for ANAs had the speckled pattern and 11%—the homogeneous one [149]. In an Australian study, when ANAs were detected above the cutoff titer in healthy children, speckled, nucleolar, homogeneous, and mixed types of fluorescence were arranged in decreasing order of frequency [148]. In healthy Mexican children, positive for ANAs, the speckled pattern was found in 50% of cases, homogeneous—in 44%, and nucleolar—in 6% [152]. The results of ANA testing in healthy children in Thailand were somewhat different: when positive results were detected, the homogeneous pattern was determined in 47% of patients, speckled—in 20%, and nucleolar—in 10% [188].

The results of studies involving healthy adults are similar to those obtained in children. Mexican [20] and Brazilian scientists [164] reported that about half of healthy subjects with positive results of ANA tests had a speckled type of fluorescence. When examining healthy subjects in the USA, speckled and homogeneous patterns were found in 84.6% of cases [21]. In our study, when positive results of ANA tests were found in healthy Russians, the most common pattern was speckled (more than 60% of positive results), followed in descending order by homogeneous, nucleolar, cytoplasmic, and mixed ones [185].

It should be noted that the speckled and homogeneous patterns are the most common ones also in rheumatic ADs [16]. Both these patterns may be due to the presence of antibodies to multiple antigens and are not specific for any particular disease. All this makes the diagnostic process even more difficult because the type of staining pattern cannot be a specific criterion for distinguishing between sick and healthy among the examined persons. Therefore, the reasonable determination of the cutoff for the ANA titer is the most important.

In recent years, special attention has been paid to the AC-2 or DFS (dense fine speckled) pattern, which is a subtype of speckled sub-pattern, commonly detected in healthy individuals (especially in young adults) who are positive for ANAs [12,16,23,57,58,101,164,189].

This is a fluorescence of interphase nuclei in the form of small dense specks, with a fine-speckled fluorescence of chromosomal regions (but not nucleolar ones) in metaphase cells. It is associated with autoantibodies against the protein DFS70 (see above). The antigen is lens epithelium-derived growth factor or transcription co-activator with a molecular weight of 75 kD.

This transcriptional co-regulator produces an important signal, activating a number of intracellular mechanisms, non-specifically increasing the survival of various cells in response to damaging influences.

The AC-2 pattern is rarely found in persons who develop rheumatic AD (0.5–3%), but in healthy people it is present more often (6–11%) [16,57,58].

Currently, there is evidence that this pattern, when confirming the presence of anti-DFS-70 autoantibodies by immunoassay methods, can serve as a biomarker for the exclusion of systemic rheumatic AD with borderline ANA titers [12,16,23,57,58,101,164,189]. It is possible that these autoantibodies are a characteristic manifestation of “beneficial autoimmunity” (see above) [61]. However, according to the ICAP, they can be considered an “anti-marker” only in the absence of other autoantibodies associated with systemic rheumatic AD [16]. In individuals with the AC-2 pattern it is possible to identify autoantibodies typical for the specific diseases by immunoassay methods and to establish the corresponding diagnoses in about 11–12% of cases [194,195]. Therefore, the presence of the AC-2 pattern is not an “indulgence” of rheumatological health, but a sign that requires clarifying tests by means of other laboratory methods.

Comparing the cytokine profiles and immune cell subtypes of healthy subjects and SLE patients positive for ANA, the American authors found that healthy individuals had significantly lower levels of several endogenous adjuvant-like cytokines (interferons, B-lymphocyte stimulation factor BlyS). Both pro-inflammatory IL-12 and stem cell factor c-Ki levels were significantly lower in healthy people, while the levels of the anti-inflammatory IL-1 receptor antagonist—IL-1Ra, on the contrary, turned out to be lower in SLE patients [196]. This corresponds to the danger model and the concept of adjuvant-like effects in defining the line between physiological and pathological autoimmunity (see above). The same research group recently highlighted the importance of race in early autoimmune profiles and identified a novel immune endotype with hallmarks of suppression in European American ANA(+) healthy individuals [197]. This immune endotype includes lower expression of T-cell activation markers, lower plasma levels of IL-6, reduced numbers of T cells, NK cells, and autoimmunity-associated B cells. Regarding the mechanism underlying the transition from an ANA(+) healthy status to SLE, the authors reported T-cell immune suppression signature in ANA(+) healthy individuals characterized by the marked downregulation of interferon-inducible genes and HLA class I genes in T cells of these individuals compared with ANA(−) controls and patients with SLE. It was suggested that suppression may be a form of regulation in response to early autoreactivity or a pathogenic result of unseen immune activation in European American ANA(+) healthy individuals.

Thus, for the correct interpretation of ANA tests, not only the ANA titer, but also the presence or absence of certain qualitative features of the HEp-2 IFA staining pattern, the presence of “protective” anti-DFS70 autoantibodies, and the features of some other parameters of the immune reactivity should be used.

Whether anti-DFS70 autoantibodies, in addition to the properties of an “anti-marker” of systemic autoimmune rheumatic pathology, represents a biomarker for the presence of any other AD is unclear, although they are often detected in Vogt-Koyanagi-Harada uveomeningitis, which occurs with frequent systemic lesions, in chronic fatigue syndrome/myalgic encephalomyelitis, in atopic dermatitis and interstitial cystitis, less often in Hashimoto’s thyroiditis, alopecia areata, sarcoidosis, and paraneoplastic phenomena [195,198]. According to Ochs et al. [198], at least 5–10% of healthy people who do not develop rheumatic diseases during follow-up can be anti-DFS70 carriers. It has been recently shown that an anti-DFS70 antibody test could help to avoid unnecessary follow-up diagnostic procedures in ANA-positive subjects with undifferentiated features of systemic autoimmune disease and minimize the use of health resources [199].

Since the LEDGFp75 autoantigen, which is the target of these antibodies, is involved in the activation of nonspecific cell responses to damage and coordination of some mechanisms that ensure cell survival, it can be suggested that the generation of anti-DFS70 antibodies may reflect the physiological autoimmunity response to the hyperexpression of LEDGF antigen by the cells exposed to damaging agents—as postulated by the theory of immunological clearance described above [41,45]. These antibodies can also arise by the idiotype-anti-idiotypic mechanism, in response to the generation of antibodies against pathogenic factors that address the lens epithelial growth factor receptor [36,60].

In a recent study from Italy, treatment of rheumatoid arthritis with a TNFα blocker was associated with the increased expression of anti-DFS70 antibodies, which is an additional argument in favor of their sanogenic rather than pathogenic role [200].

While, from the fundamental standpoint, these autoantibodies are one piece of the evidence for the existence of physiological autoimmunity, but from a clinical point of view they remain a mysterious tile in the ANA mosaic. Recently established antigen knockout DFS70 HEp-2 cells for ANA testing will contribute to the development of the approach to the differential diagnosis between physiologic and pathological autoimmunity. It is also necessary to consider the epitope differences of the LEDGFp75 domains used in immunoassay methods designed for the detection of these autoantibodies [58,187]. As for the biological significance of anti-DFS-70 autoantibodies, we cannot determine this without direct experimental study of their effect on living cells and laboratory animals.

## 5. Conclusions

To summarize, ANAs can be detected in healthy adults, adolescents, and children. This should be interpreted considering concepts of physiological autoimmunity as well as natural regulatory and functional autoantibodies. Most often in healthy individuals, ANAs are found at low titers, which makes it important to revise the cutoff for ANA testing. It is advisable to consider ANAs at titers 1/320 or greater as positive in the diagnostic process. It is of great importance to take into account the HEp-2 IFA staining patterns, especially the AC-2 pattern, in view of its role as an anti-marker of rheumatic pathology.

There are geo-epidemiological differences in ANA prevalence among healthy individuals. The reasons may be the genetic characteristics of the population, the level of environmental pollution and degree of urbanization, natural and geographical factors, the standards of living, and health care in particular areas; however, there is no complete parallelism between the detection of ANAs in healthy people and the regional prevalence of ADs.

In clinical practice, it is necessary to verify the cutoff for the ANA titer in each particular country or region and rationally combine in practical algorithms both data obtained by IFA and solid-phase assays. The first steps to suggest such an algorithm were already performed recently [201,202,203]. Further development of international documents and recommendations covering this area [204] is relevant. Such recommendations should consider regional and age-related population studies and should also be based on modern achievements in biomedical science and a paradigm shift in the question of the existence of physiologic autoimmunity.

## Figures and Tables

**Figure 1 antibodies-10-00009-f001:**
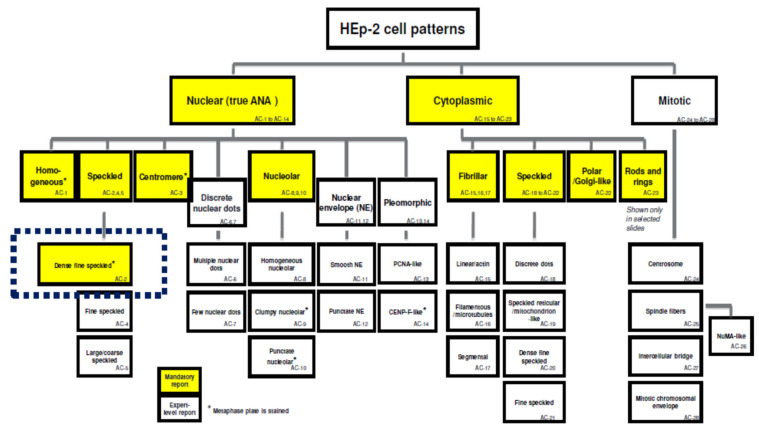
Classification of the main patterns of fluorescence in HEp-2 IFA. On a yellow background, there are patterns mandatory for registration in routine laboratory practice, on a white background—those verified by experts. * metaphase plate is stained (from: [189]).

**Table 1 antibodies-10-00009-t001:** Prevalence of ANAs among healthy adults depending on the cut-off titer.

Author(s), Year, (Reference)	Country(ies)	Age, Years	n	Share of ANA Positive Depending on Titers, %					
				1/40	1/80	1/100	1/160	≥1/320	Total
Tan E.M. et al., 1997 [165]	International (USA, Europe, Australia, Canada, Japan)	21–60	125	31.7	1.3	N/A	5.0	3.3	41.3
Fernandez S. et al., 2003 [97]	Brazil	18–60	500	14.6	4.6	N/A	2.0	1,4	22.6
Cacciapaglia F. et al., 2008 [166]	Italy (Filipinos)	25–65	80	N/A	N/A	23.7	N/A	N/A	23.7
	Italy (Italians)	25–69	60	N/A	N/A	8.3	N/A	N/A	8.3
Marin G.G. et al., 2009 [20]	Mexico	12–72	304	35.4	13.4	N/A	3.2	1.6	53.6
Mariz H. et al., 2011 [164]	Brazil	18–66	918	N/A	5.9	N/A	1.0	5.9	12.9
Satoh M. et al., 2012 [21]	USA	20–29	686	N/A	13.1	N/A	N/A	N/A	13.1
		30–39	642	N/A	13.4	N/A	N/A	N/A	13.4
		40–49	581	N/A	11.5	N/A	N/A	N/A	11.5
		50–59	478	N/A	17.4	N/A	N/A	N/A	17.4
		60–69	525	N/A	13.8	N/A	N/A	N/A	13.8
		>70	625	N/A	19.2	N/A	N/A	N/A	19.2
Racoubian E. et al., 2016 [167]	Lebanon	<20–>70	10,814	N/A	N/A	20.0	3.7	2.8	26.5
Morawiec-Szymonik E. et al., 2020 [127]	Poland	18–>60	41	N/A	N/A	N/A	N/A	N/A	4.9

**Table 2 antibodies-10-00009-t002:** Share of ANA-test positive cases among healthy children and adolescents (N/A—for absence of data).

Author(s), Year (Reference)	Country(ies)	Age, mo/Years	n	Share of ANA Positive Depending on Titers, %					
				1/40	1/80	1/160	≥1/320	≥1/640	Total
Arroyave C. et al., 1988 [184]	USA	4 months—16 years	241	0.4	N/A	N/A	N/A	N/A	0.4
Allen R.C. et al., 1991 [148]	Australia	1–16 years	100	9.0	N/A	7.0	N/A	2.0	18.0
Hilário M.O. et al., 2004 [149]	Brazil	6 months–5 years	63	N/A	3.2	1.5	1.5	1.5	8.0
		5–10 years	77	N/A	9.1	5.2	2.6	2.6	19.5
		10–15 years	49	N/A	2.0	4.0	2.0	2.0	10.0
		15–20 years	25	N/A	0.0	0.0	8.0	0.0	8.0
Wananukul S.et al., 2005 [188]	Thailand	7–15 years	207	9.6	2.9	2.9	0.0	0.0	15.4
Satoh M. et al., 2012 [21]	USA	12–19 years	1190	N/A	11.2	N/A	N/A	N/A	11.2
Somers E.C. et al., 2017 [152]	Mexico	9–17 years	114	N/A	5.3	3.5	7.0	N/A	15.8
Attilakos A. et al., 2020 [129]	Greece	4–14 years	40	N/A	N/A	5.0	N/A	N/A	5.0

## Data Availability

Data sharing not applicable.

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
