# Peer review of "Antinuclear Autoantibodies in Health: Autoimmunity Is Not a Synonym of Autoimmune Disease"

_2073-4468, 2021, doi:10.3390/antib10010009_

Round 1

Reviewer 1 Report

  1. It would preferably to limit literature sources to five - seven years.
  2. The second chapter needs to be edited both in terms of describing natural antibodies (function in norm and pathology) and at the end of the chapter to give a more focused description of literature regarding “physiological autoantibodies” and “physiological autoimmunity” as well as their opinion on this issue.
  3. When characterizing anti DFS 70 antibodies, it is preferable to add the latest publications:

Zheng B, et al Anti-DFS70 antibodies among patient and healthy population cohorts in China: Results From a multicenter training program showing spontaneous abortion and pediatric systemic autoimmune rheumatic diseases are common in anti-DFS70 positive patients. Front. Immunol. 11:562138. doi: 10.3389/fimmu.2020.562138

Luca Moroni, et al Economic Analysis of the Use of Anti-DFS70 Antibody Test in Patients with Undifferentiated Systemic Autoimmune Disease Symptoms. The Journal of Rheumatology August 2020, 47 (8) 1275-1284; DOI: https://doi.org/10.3899/jrheum.19053 

  1. Conticini et al, Anti-dense fine speckled 70 antibodies in primary Sjögren’s syndrome Clinical and Experimental Rheumatology 2020, S-326

4, It will be nice to correct the English again

Author Response

Thank you for your letter and for the opportunity to revise our paper “Antinuclear autoantibodies in health: autoimmunity is not a synonym of autoimmune disease”. We appreciate your insightful comments. They were incorporated into the revised version of the manuscript. We hope that you will find the corrected version feasible for publication in Antibodies.

Point 1: It would preferably to limit literature sources to five - seven years.

Response 1:

Perhaps for utilitarian purposes this should be done in the way recommended by a reviewer. However, the article is not limited to considering narrowly methodological issues of the practice of determining antinuclear autoantibodies. It sets a conceptual goal:  To reveal a non-trivial approach based on the concept of autoantibodies as physiological bioregulators. Such a goal can be achieved problem can only by referring to the history of ideas. Otherwise, the reader will remain constrained by a common knowledge or momentarily fashionable thesaurus. It is not possible to understand a given result without clarifying its path of development that led to the given result. Authors stand on the position of the unity of science, both logical and historical approaches in comprehension of a problem.

Point 2: The second chapter needs to be edited both in terms of describing natural antibodies (function in norm and pathology) and at the end of the chapter to give a more focused description of literature regarding “physiological autoantibodies” and “physiological autoimmunity” as well as their opinion on this issue.

Response 2:

We are in complete agreement with the reviewer about the meaning of Chapter 2 and the desirability of covering as much detail as possible concerning the most nontrivial of the concepts involved - namely, the concept of physiological autoimmunity. Therefore, we have revised this chapter in order to improve its perception by the readers. But the article is already voluminous, which, in our opinion, does not provide an opportunity to expand this section and retell the original sources, which are cited and available to the readers for additional information.

The authors understand that the detailed content of the cited literature containing unorthodox ideas about the role of autoimmunity may be of great interest to readers. Also, the authors understand that there are readers who speak only English among the main UN languages, and previously did not pay attention to those primary sources cited in the article, which were published outside the English-speaking part of the world. But this is still a review, not a reproduction or translation of the original references, therefore, respecting the advice of the reviewer, the authors did not dare to expand further on Chapter 2. The main goal was to draw attention of readers to the concept, all facets can be obtained, of course, by detailed reading of cited sources.

Point 3: When characterizing anti DFS 70 antibodies, it is preferable to add the latest publications:

Zheng B, et al Anti-DFS70 antibodies among patient and healthy population cohorts in China: Results From a multicenter training program showing spontaneous abortion and pediatric systemic autoimmune rheumatic diseases are common in anti-DFS70 positive patients. Front. Immunol. 11:562138. doi: 10.3389/fimmu.2020.562138

Luca Moroni, et al Economic Analysis of the Use of Anti-DFS70 Antibody Test in Patients with Undifferentiated Systemic Autoimmune Disease Symptoms. The Journal of Rheumatology August 2020, 47 (8) 1275-1284; DOI: https://doi.org/10.3899/jrheum.19053 

Conticini et al, Anti-dense fine speckled 70 antibodies in primary Sjögren’s syndrome Clinical and Experimental Rheumatology 2020, S-326

Response 3: We agree with the Reviewer and added the suggested by the Reviewer publications (see in the revised version of the manuscript:

“..This was demonstrated both for the autoantibodies to cell surface receptors [42, 53-56] and for ANA [12, 57]. For example anti-DFS70 autoantibodies against the lens epithelium-derived growth factor are more prevalent in healthy people than in patients with ADs, and have been considered in recent years as an important marker of the lower probability of rheumatological diseases, even when ANA test is positive [58]. However, recent findings raise some questions regarding clinical relevance of anti-DFS70 autoantibodies. Zheng et al. showed that frequency of systemic autoimmune rheumatic diseases in anti-DFS70 positive pediatric patients was unexpectedly as high as 50.0% [59]. Conticini et al. reported that 7/9 (77.8%) anti-DFS70 positive adult patients with clinical suspicion of primary Sjogren syndrome received this diagnosis after minor salivary glands biopsy [60]..

.. According to Ochs et al. [199], at least 5-10% of healthy people who do not develop rheumatic diseases during follow-up can be anti-DFS70 carriers. It has been recently shown that anti-DFS70 antibody test could help to avoid unnecessary followup diagnostic procedures in ANA-positive subjects with undifferentiated features of systemic autoimmune disease and minimize the use of health resources [200]..”

Point 4: It will be nice to correct the English again

Response 4: We agree with the Reviewer and corrected the English (see in the revised manuscript)

Reviewer 2 Report

  1. The authors have made a heroic effort to justify the title of this review with considerable success. References are extensive, from William Shakespeare and Elie Metchnikoff to mid-20th century researchers and publications up to the current year. This no small undertaking has produced a thorough treatise, which deserves praise and publication.
  2. A conceptual criticism: Much of the theoretical aspects of this commentary focus on autoantibodies as if they are autonomous products of B cell output. Only do we get an passing acknowledgment of the importance of T cells in providing the critical component in production of high titer and high affinity potentially pathological autoantibody production in Line 204 with the statement “The process of antigen presentation and the effects of T cells on this process can determine, therefore, whether autoimmunity will be maintained within the physiological limits, or it will evolve in AD.” Yet the authors devote Section 2 to an extensive discussion of older ideas in the literature that autoantibodies in healthy individuals, and presumably all vertebrates, are a manifestation of some important physiological function, such as immunological clearance of cellular debris, idiotype/anti-idiotype regulation, or even a >100 year-old idea on a role of the immune system in “…forming and maintaining the multicellularity of the metazoan organism”.   This latter dubious attribution to an otherwise great Russian biologist (Metchnikoff) as well as the other attempts to argue for an important physiological role of (low titer?) autoantibodies ignores the fact that the vast majority of multicellular organisms (invertebrates) lack such molecules since they do not have an adaptive immune system. In the end the simplest explanation of the common presence of low titer autoantibodies – that they are products of autoreactive B cells that fail to receive “help” from autoreactive T cells – is not given any real consideration in this discussion. This oversight detracts from the value of Section 2.
  3. English problems throughout. A few examples of problems in grammar, spelling, syntax, punctuation:
  4. Line 53, “there is a growing number of evidence; line 72, “biological secrets”; line 133 and elsewhere, “1980ies”; line 268, “Oochterlony's test”; line 298, “a significant proportion of ADs belongs”; line 330, “with the fluorescence microscopy”
  5. Excessive and inappropriate use of exclamation points.
  6. Excessive paragraphing greatly detracts from the points the authors are trying to make and the overall readability of this document.
  7. Lines 407-410: largely incomprehensible; “Sections 1-2” does not exist.
  8. Table 1 needs a title
  9. Lines 450-451. This statement is quite a leap; maybe the European medical community just prefers an ANA test that is more diagnostically useful.
  10. Possible errors. Lines 460-461. Do the authors mean to say “…fluorescence at a dilution of 1/20 was not found determined and therefore reported as negative in the first second laboratory”; Line 536, “AHA-positivity” = ANA-positivity; Line 602 also reads “AHA” not ANA

Author Response

Thank you for your letter and for the opportunity to revise our paper “Antinuclear autoantibodies in health: autoimmunity is not a synonym of autoimmune disease”. We appreciate your insightful comments. They were incorporated into the revised version of the manuscript. We hope that you will find the corrected version feasible for publication in Antibodies.

Point 1: A conceptual criticism: Much of the theoretical aspects of this commentary focus on autoantibodies as if they are autonomous products of B cell output. Only do we get an passing acknowledgment of the importance of T cells in providing the critical component in production of high titer and high affinity potentially pathological autoantibody production in Line 204 with the statement “The process of antigen presentation and the effects of T cells on this process can determine, therefore, whether autoimmunity will be maintained within the physiological limits, or it will evolve in AD.” Yet the authors devote Section 2 to an extensive discussion of older ideas in the literature that autoantibodies in healthy individuals, and presumably all vertebrates, are a manifestation of some important physiological function, such as immunological clearance of cellular debris, idiotype/anti-idiotype regulation, or even a >100 year-old idea on a role of the immune system in “…forming and maintaining the multicellularity of the metazoan organism”.   This latter dubious attribution to an otherwise great Russian biologist (Metchnikoff) as well as the other attempts to argue for an important physiological role of (low titer?) autoantibodies ignores the fact that the vast majority of multicellular organisms (invertebrates) lack such molecules since they do not have an adaptive immune system. In the end the simplest explanation of the common presence of low titer autoantibodies – that they are products of autoreactive B cells that fail to receive “help” from autoreactive T cells – is not given any real consideration in this discussion. This oversight detracts from the value of Section 2.

Response 1:

Authors agree with the generally accepted view of the reviewer that T-lymphocytes play an important role in controlling the spectrum and intensity of autoimmune responses. It would be possible to highlight this in more detail, as well as to touch upon the topic of the role that B-regs and other mechanisms play in these processes. But the authors created their article for the journal “Antibodies”, and therefore consider the emphasis on the role of autoantibodies as regulators appropriate and necessary.

Authors do not see anything inappropriate in discussing the modern development of ideas that appeared more than 100 years ago. The theory of relativity appeared more than 100 years ago, and the corresponding quotations certainly do not apply to the last 5-7 years - and nevertheless, in theoretical Physics, these sources and roots are addressed and cited. Why not do the same in an article treating the issues of theoretical Immunobiology? Accordingly, the reference to Mechnikov, in our opinion, is not only appropriate, but necessary. The reviewer, as can be judged from the context of his comments, is not an adherent of the concept of physiological autoimmunity. But this does not mean that his position is newer than the position of the authors of the article. In fact, Paul Ehrlich, Mechchkov's friend and opponent, spoke with the same logic and ideas more than 100 years ago (1900). And he also considered Mechnikov's idea "dubious". The opposition of horror autotoxicus and physiological inflammation in Medicine is like the lines of Plato and Democrites in Philosophy – will exist continuously in spite of fashions and new methods.

Mechnikov in 1882 predicted that  immune system’s prior task in norm is control of ontogenesis and support of multicellularity (morphogenetic function) realized by physiologic inflammation (autoimmunity). Nüsslein-Volhard 100 years later discovered homeosis genes of morphogenesis and their products which are able to self-assembly and are recognized by Toll-receptors of Invertebrates, which process controls Drosophilae embiogenesis.  Jules Hoffmann showed that Toll-like receptors and their ligands involved not only in embryogenesis but also in innate immunity of insects Both are Nobelians (1995 & 2011).  But, these or similar (Toll-like) receptors and absolutely homologous self-assembling effectors turned out to control innate immunity and its relations to adaptive immunity – not only in Invertebrates, but also in Mammals.

If the same of homologous molecules are controlling embryo-genesis and immunity – does it mean that Mechnikov was dubious? Of course, in real science everything is dubious, but to our mind all these recent achievements stand in line with Mechnikov’s idea.

A reviewer's observation that Invertebrates lacked adaptive immunity 10 years ago would have been compelling ultimate truth. But now, when it has been firmly proven that adaptive immunity is developed even in Bacteria and Archaea - the authors mean the existence of CRISPR-Cas mechanisms that are precisely adaptive immunity - such a categoricalness in this matter looks archaic. Invertebrates have autophagocytosis, it plays a morphogenetic role in them, establishes the reduction of provisional organs, and relies upon recognition molecules, the homologues of which are both antibodies and receptors of the vertebrate immune system.

Hence, absence of the lymphoid organs in Invertebrates does not make the concept which we defend obsolete.

Next, it seems to us that the reviewer has in vain put a question mark after citing our thesis that antibodies in low titers can be regulators, and in high concentrations, vice versa instruments of damage.

This is a general biological law. No one is surprised, for example, that such a molecule recognized by receptors as vasopressin regulates osmotic homeostasis and blood pressure in a certain concentration range, but when its  blood concentrations in hemodynamic shock and severe combined traumas increase 1000 times - it is vasopressin, and not some other, unusual for the normal conditions molecule, that serves as the main mediator of disseminated intravascular coagulation. Quantity turns into quality.

 Of course, the low titers of natural autoantibodies are derived from B-lymphocytes not getting Th help – and we do not stand it under doubt in any portion of our text.

 But this mechanism of their generation does not rule out their possible regulatory function, and does not contradict to the concept of autoantibody-mediated regulation.  They can recognize, they can bind – and citing Paul Ehrlich we may say “Corpora non facit nisi fixata”.

Of course natural autoantibodies are generated by the mechanism mentioned by reviewer. But the point is if they are occasional by-products valid for nothing or have certain function, like bilirubin, CO2, uric acid, trace amines – which are not just garbage, but all have homeostatic roles.

   Thus, epistemologically, the argument given by the reviewer does not serve to deny the main idea of Chapter 2. The reviewer refutes not what the authors wanted to emphasize, he argues not with the author's, but with his own perception of the concept.

Nevertheless, for objectiveness we feel it appropriate to follow the advise given by respected reviewer and include into the text the simpliest explanation which he recommended (after the line 184.

“Low titer autoantibodies in healthy organism, most probably, are products of autoreactive B cells that fail to receive “help” from autoreactive T cells. But this is just the mechanism of their generation, it does not rule out the possibility of their regulatory effects in health, because these molecules are able to recognize and bind their targets altering their metabolic destiny and/or lifecycle.”.

Point 2: Line 53, “there is a growing number of evidence; line 72, “biological secrets”; line 133 and elsewhere, “1980ies”; line 268, “Oochterlony's test”; line 298, “a significant proportion of ADs belongs”; line 330, “with the fluorescence microscopy”

Response 2: We agree with the Reviewer and corrected these problems: “there is a growing body of evidence”; “mucous secretions”; “1980s”; “Ouchterlony test”; “a significant proportion of ADs belong”; “with the fluorescence microscope”.

Point 3: Excessive and inappropriate use of exclamation points.

Response 3: We agree with the Reviewer and removed almost all exclamation points.

Point 4: Excessive paragraphing greatly detracts from the points the authors are trying to make and the overall readability of this document.

Response 4:

Paragraphing is not purely author’s  but also an editorial matter. Authors do not insist on the version of paragraphing used in their manuscript. In normal editorial process those who prepare a layout  of a book or journal - are free to adjust paraghraphing to the needs of layout optimization. Anyway, it can be corrected at the step of galley proof – accordingly by technical editor and author.

Point 5: Lines 407-410: largely incomprehensible; “Sections 1-2” does not exist.

Response 5: In these lines we refer to the Section 1 ”Introduction” and Section 2 “Physiological autoimmunity and its bidirectional pathological changes” where we discuss physiological autoimmunity and natural autoantibodies. In the revised manuscript we made it clearer:

“In view of the above (see Sections 1 and 2), this approach looks like not sufficient and not up to date.”

Point 6: Table 1 needs a title

Response 6: We agree with the Reviewer and added a title for the Table 1

“Table 1 Prevalence of ANA among healthy adults depending on the cut-off titer.”

Point 7: Lines 450-451. This statement is quite a leap; maybe the European medical community just prefers an ANA test that is more diagnostically useful.

Response 7:

Authors are agree with a reviewer, and omitted this statement from the text

Point 8: Possible errors. Lines 460-461. Do the authors mean to say “…fluorescence at a dilution of 1/20 was not found determined and therefore reported as negative in the first second laboratory”; Line 536, “AHA-positivity” = ANA-positivity; Line 602 also reads “AHA” not ANA

Response 8: We agree with the Reviewer. These are errors. We have corrected them. See in the revised manuscript.

“Since 1/40 and 1/20 titers were used as diagnostic in the first and in the second laboratories respectively, fluorescence at a dilution of 1/20 was not found  and therefore reported as negative in the first laboratory.”

“ANA-positivity in the Americas, according to literature, is higher than in Eurasia. Thus, out of 304 healthy Mexicans, 17.9% had ANA at a titer of 1/80 and more, but only 1.3% of these individuals were positive at 1/320 titer”

“But there is no clear parallelism between the distribution of the ADs and the regional prevalence of ANA in healthy individuals, apparently due to the incomplete adequacy of interlaboratory comparisons.”

Reviewer 3 Report

The authors present a summary on the role of ANA in physiological autoimmunity and pathological processes.  They analyzed several aspects including ANA detection, polyspecificity, relation to the ADs pathogenesis and detection in healthy individuals.

Overall, the manuscript does not flow properly, chapter are really long, and the story could be shortened to have a greater impact on the key topic. The presence of subchapters could help to better understand the point. On the other hand, the descriptions of ANA in the introductory section is really short.

The review should be organised in a better way, it is difficult to clearly understand the point the authors are trying to make.

Specific points:

-Please explain better the presence of ANA in healthy and the relevance for the immune regulation.

- Primary methods available to clinical laboratories as screening for ANA should described in a better way. Additionally, a table summarizing it should be presented. It can be of interest for the readers

- Immunological difference between ANA + healthy individuals and patients should be described.

- The text presents several repetitions: e.g. lines 144 and 151

- A figure describing diseases in which ANA are often identified should be included. Additionally, I think that few words describing ANA production can be of interest for the readers. 

- Table 1, title is missed

- “Regional, social, and racial-ethnic aspects of ANA prevalence” the chapter is really long

- Slight-Webb and colleagues, (2020, 10.1016 / j.jaci.2020.04.047) recently determine the immune features that might define and prevent transition to clinical autoimmunity in ANA + healthy individuals. Please comment it in your manuscript.

- Typo and grammar errors should be corrected.

Author Response

Thank you for your letter and for the opportunity to revise our paper “Antinuclear autoantibodies in health: autoimmunity is not a synonym of autoimmune disease”. We appreciate your insightful comments. They were incorporated into the revised version of the manuscript. We hope that you will find the corrected version feasible for publication in Antibodies.

Point 1: Overall, the manuscript does not flow properly, chapter are really long, and the story could be shortened to have a greater impact on the key topic. The presence of subchapters could help to better understand the point. On the other hand, the descriptions of ANA in the introductory section is really short.

Response 1: Considering that one of the respected reviewers complains about the too detailed division of the article into paragraphs, and the other, on the other hand, advises to divide it more fractionally, then one must admit, that the authors, apparently, fell just between these extreme points of view - that is, at the optimum. Truly. how many people, so many opinions.

Nevertheless, the authors, respecting the opinion of the reviewer 3, introduce subsections in the longest chapter 2.

The introduction solves the introductory problem and cannot provide detailed descriptions. In the further presentation of the ANA, they are far from briefly described, many details are given, which the reviewer himself noted above.

To make long story short is not a task for a conceptual review of a contradictory topic.

One of reviewers vice versa insists on necessity of adding brief content of some references given, so authors again feel themselves in between the extreme recommendations to broaden and to shorten, which means they stand close to optimum.

If the space economy is critical for the journal, authors are agree to omit figure 1

Point 2: The review should be organised in a better way, it is difficult to clearly understand the point the authors are trying to make.

Response 2:  Perhaps other authors would organize it in some other way. We did our best and reading the reviews we noticed that other reviewers did not complain that our text is not clear or the message is not understandable. But tests differ. On the contrary, it seemed to some of the reviewers that the main idea of the article on the physiological regulatory potential of natural autoantibodies was expressed too clearly, with exclamation marks.

However, respecting the preferences and habits of those respected colleagues, who like to keep everything in brief - we, in fact, put the main message into the paper’s headline. Autoimmunity is not a disease yet. You can't put it more shortly.

Point 3: Please explain better the presence of ANA in healthy and the relevance for the immune regulation.

Response 3: This is still a matter of debates, that’s why we already dedicated to historical and current explanations of the physiological autoimmunity the whole chapter (#2), longest in the text.

Authors do not feel that it is possible to satisfy both point 2 (to be short) and point 4 (to broaden) out of the list of notes given by a reviewer.

Point 4: Primary methods available to clinical laboratories as screening for ANA should described in a better way. Additionally, a table summarizing it should be presented. It can be of interest for the readers

Response 4: We could write in more detail about the ANA research methods, but the paper is not methodological guide, rather it refers to several references which contain the methodological details. Sapientis sat. We do not describe methods, but only give a brief idea of the problematic moments in the interpretation of data associated with changing fundamental paradygms about the role of autoimmunity in the body and about the physiological significance of autoantibodies.

It is inappropriate to cite other methods, since our material was 95% devoted to the study of the ANA, it was even included in the title. Moreover, it is redundant to make a comparative table by methods.

Point 5: Immunological difference between ANA + healthy individuals and patients should be described.

The main difference between them is health or disease

Response 5:

Point 6: The text presents several repetitions: e.g. lines 144 and 151

Response 6: We agree with the Reviewer and removed the repetition. See in the revised manuscript:

Natural autoantibodies can also belong to immunoglobulin isotypes other than IgM [52].”

Point 7: A figure describing diseases in which ANA are often identified should be included. Additionally, I think that few words describing ANA production can be of interest for the readers. 

Response 7: The article in general is devoted to healthy people and to the role of anti-nuclear autoantibodies in the norm. Therefore, the introduction of a table describing the clinical significance of ANA for different diagnoses seems to us superfluous. Moreover, on pages 21-22)- we  provide links to sources that cover in detail the modern point of view on the clinical significance of different ANA patterns (16,  140,188, 189, 190), and even briefly give the main correlates. All the latest consensus sources emphasize a departure from the perception of certain ANA patterns as markers of certain diagnoses, insisting on a broader wording of «clinical relevance”. Hence, in our paper we did not dare “to be more holy than the Pope of Romans”

Point 8: Table 1, title is missed

Response 8: We agree with the Reviewer and added a title for the Table 1

“Table 1 Prevalence of ANA among healthy adults depending on the cut-off titer.”

Point 9: “Regional, social, and racial-ethnic aspects of ANA prevalence” the chapter is really long

Response 9: Authors do not think that 2 pages (out of 35 of total length or 6% is really too long.  The fact that individuals are biologically different in different regions and communities is practically very important

Point 10: Slight-Webb and colleagues, (2020, 10.1016 / j.jaci.2020.04.047) recently determine the immune features that might define and prevent transition to clinical autoimmunity in ANA + healthy individuals. Please comment it in your manuscript.

Response 10: We agree with the Reviewer and comment the paper of Slight-Webb et al. in our manuscript. See in the revised version:

“Comparing the cytokine profiles and immune cell subtypes of healthy subjects and SLE patients positive for ANA, the American authors found that healthy individuals had significantly lower levels of several endogenous adjuvant-like cytokines (interferons, B-lymphocyte stimulation factor BlyS). Both pro-inflammatory IL-12 and stem cell factor c-Ki levels were significantly lower in healthy people, while the levels of the anti-inflammatory IL-1 receptor antagonist - IL-1Ra, on the contrary, turned out to be lower in SLE patients [197]. This corresponds to the danger model and the concept of adjuvant-like effects in defining the line between physiological and pathological autoimmunity (see above). The same research group have recently highlighted the importance of race on early autoimmune profiles and identified a novel immune endotype with hallmarks of suppression in European American ANA(+) healthy individuals [198]. This immune endotype includes lower expression of T-cell activation markers, lower plasma levels of IL-6, reduced numbers of T cells, NK cells, and autoimmunity-associated B cells. Regarding the mechanism underlying the transition from an ANA(+) healthy status to SLE, the authors reported T-cell immune suppression signature in ANA(+) healthy individuals characterized by the marked downregulation of interferon-inducible genes and HLA class I genes in T cells of these individuals compared with ANA(-) controls and patients with SLE. It was suggested that suppression may be a form of regulation in response to early autoreactivity or a pathogenic result of unseen immune activation in European American ANA(+) healthy individuals.

Point 11: Typo and grammar errors should be corrected.

Response 11: We agree with the Reviewer. The manuscript has been checked again. Typo and grammar errors were corrected. See in the revised version of the manuscript.

Round 2

Reviewer 3 Report

I carefully reviewed the revised manuscript "Antinuclear autoantibodies in health: autoimmunity is not a synonym of autoimmune disease" and the responses from the authors and there are still many concerns that the authors did not respond sufficiently. I think that the quality of the article has not improved significantly.

Author Response

Reply to the Report of Reviewer 3 «I carefully reviewed the revised manuscript "Antinuclear autoantibodies in health: autoimmunity is not a synonym of autoimmune disease" and the responses from the authors and there are still many concerns that the authors did not respond sufficiently. I think that the quality of the article has not improved significantly».

With all respect to the views and opinion of reviewer â„–3 (even contradicting or disagreed with the opinions of other two reviewers and authors), we shall repeat that  to make long story short is not a task for a conceptual review of a contradictory topic, which already demonstrated its sensitive, meaningful  and even a bit provoking character – as regards to the  commentaries of the reviewer â„–3.

We noticed that other reviewer vice versa insisted on necessity of adding brief content of some references given, so authors feel themselves in between the extreme recommendations to broaden and to shorten, which means they most probably stand close to optimum.

A respected colleague masked with the nickname “Reviewer â„–3” apparently is a representative of another scientific school and proponent of another paradigm than those akin to authors.

We noticed that our dear colleague relies mostly on English-medium texts and we feel lack of evidence that he is familiar with the original texts of papers cited in our review and published in other languages of continental Europe. But this is a review of world literature, not of English literature only.

With gratitude for his efforts and time spent,  we feel it appropriate to shorten our paper and we did exclude not only fig. 1, which was already done, but some more passages, wholeheartedly sharing his commendable desire for laconism  (see omitted lines 374-377, 379-381, 494-497, 563-566, 755-760, 762-763, 771-773, 784-787, 813-815...)

We have also checked grammar and language issues again (see in the revised manuscript).

 At the same time we feel that further amelioration of our text will not convert reviewer â„–3 into our proponent, hence such an amelioration seems redundant, because this is not our aim.